# Thermosensory behaviors of the free-living life stages of *Strongyloides* species support parasitism in tropical environments

Ben T. Gregory[1], Mariam Desouky[1], Jaidyn Slaughter[2], Elissa A. Hallem[3,4], Astra S. Bryant[1]*

**1** Department of Neurobiology and Biophysics, University of Washington, Seattle, Washington, United States of America, **2** BRIGHT-UP Summer Research Program, University of Washington School of Medicine, Seattle, Washington, United States of America, **3** Department of Microbiology, Immunology, and Molecular Genetics, University of California, Los Angeles, Los Angeles, California, United States of America, **4** Molecular Biology Institute, University of California, Los Angeles, Los Angeles, California, United States of America

* astrab@uw.edu

**Data Availability Statement:** All data for this study are available on GitHub: https://github.com/

## Abstract

Soil-transmitted parasitic nematodes infect over 1 billion people worldwide and are a common source of neglected disease. *Strongyloides stercoralis* is a potentially fatal skin-penetrating human parasite that is endemic to tropical and subtropical regions around the world. The complex life cycle of *Strongyloides* species is unique among human-parasitic nematodes in that it includes a single free-living generation featuring soil-dwelling, bacterivorous adults whose progeny all develop into infective larvae. The sensory behaviors that enable free-living *Strongyloides* adults to navigate and survive soil environments are unknown. *S. stercoralis* infective larvae display parasite-specific sensory-driven behaviors, including robust attraction to mammalian body heat. In contrast, the free-living model nematode *Caenorhabditis elegans* displays thermosensory behaviors that guide adult worms to stay within a physiologically permissive range of environmental temperatures. Do *S. stercoralis* and *C. elegans* free-living adults, which experience similar environmental stressors, display common thermal preferences? Here, we characterize the thermosensory behaviors of the free-living adults of *S. stercoralis* as well as those of the closely related rat parasite, *Strongyloides ratti*. We find that *Strongyloides* free-living adults are exclusively attracted to near-tropical temperatures, despite their inability to infect mammalian hosts. We further show that lifespan is shorter at higher temperatures for free-living *Strongyloides* adults, similar to the effect of temperature on *C. elegans* lifespan. However, we also find that the reproductive potential of the free-living life stage is enhanced at warmer temperatures, particularly for *S. stercoralis*. Together, our results reveal a novel role for thermotaxis to maximize the infectious capacity of obligate parasites and provide insight into the biological adaptations that may contribute to their endemicity in tropical climates.

BryantLabUW/Strongyloides-Free-Living-Thermosensory-Behaviors. The GitHub repository has also been deposited at Zenodo (https://doi.org/10.5281/zenodo.13738264).

**Funding:** This work was supported by National Institutes of Health DP2AI184544 (A.S.B.), funds provided by the University of Washington School of Medicine (A.S.B), National Institutes of Health R01AI136976 (E.A.H), and funds provided by AstraZenica and an anonymous donor (J.S.). The funders had no role in study design, data collection and analysis, decision to publish, or preparation of the manuscript.

**Competing interests:** The authors have declared that no competing interests exist.

## Author summary

Soil-transmitted parasitic nematodes infect over 1 billion people and are a major source of neglected disease, particularly in the world's most resource-limited communities. For most parasitic nematode species, reproductive adults exclusively reside within host animals. Species in the genus *Strongyloides* have a unique step in their life cycle that features soil-dwelling, non-parasitic adults. Previous studies of the free-living model nematode *Caenorhabditis elegans* have identified temperature as an important factor in the ability of free-living nematodes to survive and reproduce in the environment. Our study investigates how the thermosensory behaviors of *Strongyloides* free-living adults contribute to their survival as well as their role in amplifying the number of infective larvae in the soil. We show that *Strongyloides* free-living adults display broad thermophilic preferences that are highly distinct from those of *C. elegans* adults. We also present the first evidence that thermotaxis acts as a robust mechanism for maximizing the infectious capacity of *Strongyloides* species located in tropical climates.

## Introduction

Soil-transmitted parasitic nematodes are estimated to infect over 1 billion people worldwide and can cause a range of debilitating symptoms, including chronic gastrointestinal distress, anemia, malnutrition, and impaired growth in children [1–3]. The skin-penetrating human parasite *Strongyloides stercoralis* is estimated to infect at least 600 million people [4–7]. Chronic *S. stercoralis* infections, which arise due to the unique capacity of *S. stercoralis* to auto-infect hosts, can go undetected for many decades before evolving into a life-threatening systemic hyperinfection if an individual becomes immunocompromised [2,8,9]. Soil-transmitted parasitic worms primarily affect impoverished communities and places that lack adequate sewage management [6,7,10,11]. Current treatments only target ongoing infections and can be inadequate; no medical treatment exists to prevent infection or reinfection [12–14]. Additionally, there are currently a limited number of anthelminthic drugs, and anthelminthic resistance is already widespread among livestock parasites due to repeated dosage with anthelminthic drugs as part of regular deworming schedules [15,16]. Identifying novel therapeutic strategies, before drug resistance develops in humans, is therefore an important global health priority.

   Many soil-transmitted parasitic nematodes have a life cycle that starts with developmentally arrested third-stage infective larvae (iL3s) finding and infiltrating a host animal [17]. Once inside a host, iL3s resume development and migrate through various host tissues to the small intestine, where they take up residence as parasitic adults. Parasitic adults produce offspring that are then voided from the host in feces. These offspring develop into iL3s and restart the cycle. Species in the genus *Strongyloides* have an additional, unique step in their life cycle, in which a fraction of the offspring of parasitic adults can alternatively develop into morphologically distinct free-living adults instead of infective larvae (S1 Fig) [8,18,19]. The offspring of these free-living adults exclusively develop into iL3s that then must find and invade host animals. The molecular mechanisms that drive the choice between homogonic development (post-parasitic larvae become iL3s) and heterogonic development (post-parasitic larvae become free-living adults) is poorly understood; however, the developmental decision depends on species identity, strain genetics, host immunocompetency, and environmental temperatures [20–25]. For example, exposure to temperatures above 34°C promotes direct development into iL3s for *S. stercoralis*, presumably to allow larvae to develop into iL3s while still inside the host; these iL3s re-penetrate host tissues to establish an autoinfection [2,9,21]. The

free-living generation of *Strongyloides* spp. allows these species to sexually reproduce; unlike hookworm adults, *Strongyloides* parasitic adults are all genetically female and produce clonal progeny via parthenogenesis [8,20,26–29]. Furthermore, the *Strongyloides* free-living generation has been proposed to act as a means of amplifying the number of infective larvae that a single parasitic female can produce [30]. Despite its role in the *Strongyloides* life cycle, the *Strongyloides* free-living generation is generally understudied [31]. A better understanding of the sensory physiology and behavioral repertoires displayed by free-living *Strongyloides* adults could enable the development of novel strategies for predicting and controlling the infectious capacity of a neglected source of human disease.

The free-living adults of *Strongyloides* spp. live in the soil and experience similar environmental stressors to *Caenorhabditis elegans*, a nematode that is free-living throughout its entire life cycle. Furthermore, *Strongyloides* free-living adults share many morphological similarities with *C. elegans* adults [30,32,33]. Thus, to understand how sensory behaviors enable *Strongyloides* free-living adults to survive in the environment, we can compare their behaviors to those of *C. elegans*. One of the most ethologically significant and well-documented sensory behaviors exhibited by *C. elegans* is thermosensory navigation. *C. elegans* uses a complex set of thermosensory behaviors to stay within a physiologically and reproductively permissible temperature range (15–25°C) [34–41]. In brief, *C. elegans* will utilize both positive and negative thermotaxis to migrate towards a "remembered" cultivation temperature; at its cultivation temperature, *C. elegans* will engage in isothermal tracking [35,38]. At noxious temperatures ≥27°C and ≤5°C, *C. elegans* engage in noxious temperature responses, including reorientation behaviors and accelerated migration towards physiological temperatures [40,42]. These behavioral strategies are a critical mechanism for thermoregulation by a non-parasitic ectotherm: exposure to noxious heat stress causes dramatic drops in *C. elegans* fertility and longevity, in a manner partially dependent on strain and experience [34,43–50].

In comparison to *C. elegans*, the sensory behaviors and physiological requirements of soil-transmitted parasitic nematodes are far less documented. The few studies that have quantified the sensory behaviors of parasitic nematodes have demonstrated that the worms can detect a range of sensory cues, including chemosensory and thermosensory stimuli [51–54]. For thermosensation, the iL3s of multiple mammalian-parasitic nematode species, including *S. stercoralis*, display highly robust parasite-specific thermosensory behaviors [55–61]. These include positive thermotaxis towards mammalian body heat and negative thermotaxis towards below-ambient temperatures; the switch between these two behavioral regimes is dictated by the ambient cultivation temperature, such that worms exposed to temperatures above ambient will engage in positive thermotaxis (and vice versa) [55]. As opposed to thermosensory behaviors being primarily used for physiological thermoregulation (as with *C. elegans*), the positive thermotaxis behaviors of iL3s are thought to reflect active host seeking; negative thermotaxis behaviors are potentially a mechanism enabling iL3s to either maximize the contrast between environmental temperatures and host-emitted heat or disengage from accidental heat seeking towards a non-host heat source [55,56]. Chemosensory behaviors also contribute to host-seeking behaviors by soil-transmitted parasites [52,53,62]. Indeed, life-stage-specific chemosensory preferences have previously been proposed to contribute to the restriction of host-seeking behaviors to iL3s. For example, although *Strongyloides* iL3s and free-living adults are both attracted to a variety of host odorants, only free-living larvae and adults are attracted to fecal odors [53,62]. These findings suggest that downregulation of attraction to fecal odors enables iL3s to disperse from feces in search of hosts [53,62]. Finally, although the impact of multisensory cues on parasite behavior has not been thoroughly investigated, initial work has indicated a sensory hierarchy in which strong thermal drive can override attraction to host-associated odorants in iL3s [55]. Importantly, this hierarchy depends on the specific thermal context in

which an odorant is located. At temperatures below human skin temperature (~34˚C), the infective larvae appear to ignore attractive odorants in favor of warmer temperatures [55]. In contrast, if an odorant is placed at temperatures closer to ~34˚C, the infective larvae will decrease thermotaxis behaviors and increase local search behaviors [55]. Together, these findings suggest a sensory-dependent host-seeking strategy in which soil-transmitted iL3s first navigate towards a heat source, then determine whether that heat source is a potential host.

The thermosensory preferences of *Strongyloides* free-living adults are completely unknown. Do the physiological and behavioral responses of *Strongyloides* free-living adults to different environmental temperatures mirror those exhibited by *C. elegans* adults? To answer this question, we conducted a quantitative analysis of the thermal preferences and physiology of the free-living adults of two *Strongyloides* species: the human parasite *S. stercoralis* and the closely related rodent parasite *Strongyloides ratti*. We found that the thermosensory behaviors of *Strongyloides* free-living adults are distinct from both *Strongyloides* infective larvae and *C. elegans* adults. Across a broad range of thermal gradients, including those that trigger negative thermotaxis in iL3s and noxious heat escape in *C. elegans*, *Strongyloides* free-living adults display thermophilic migration. In addition, *S. stercoralis* free-living adults show a shifted multisensory hierarchy, such that chemosensory attractants influence behavior at temperatures below mammalian body heat. Finally, we discover that although attraction to warm temperatures shortens the lifespan of *S. stercoralis* free-living adults, their reproductive potential is enhanced. Our results suggest that *Strongyloides* free-living adults, whose progeny exclusively develop into infective larvae, use thermotaxis navigation as a mechanism to enhance the number of infective larvae in the soil. Together, these findings reveal biological adaptations that likely enhance the transmission of a potentially fatal human-parasitic nematode in tropical and subtropical climates.

## Methods

All protocols and procedures involving vertebrate animals were approved by the University of California, Los Angeles Office of Animal Research Oversight (Protocol ARC-2011-060) and by the University of Washington Office of Animal Welfare (Protocol 4570–01), which adhere to the standards of the American Association for Accreditation of Laboratory Animal Care (AAALAC) and the *Guide for the Care and Use of Laboratory Animals*.

### Maintenance of *Strongyloides stercoralis*

*S. stercoralis* UPD strain was serially passaged through male Mongolian gerbils (Charles River Laboratories) and maintained on fecal-charcoal plates as previously described [18]. Gerbils were inoculated with ~2,250 iL3s in 200 μL sterile phosphate buffered saline (PBS) via inguinal subcutaneous injection, under isoflurane anesthesia. Feces containing *S. stercoralis* were collected by placing infected gerbils on wire-bottomed cages overnight, with damp Techboard liners (Shepard Specialty Papers) on the cage bottoms. Fecal pellets were collected in the morning, mixed with autoclaved charcoal granules (Ebonex), and stored in 10-cm-diameter Petri dishes lined with Whatman filter paper. Fecal charcoal plates were stored at either 20˚C for 48 hours or 25˚C for 24 hours, then used in behavioral assays (free-living adults) or transferred to 23˚C for 5–14 days until use (iL3s). *S. stercoralis* free-living adults and iL3s were collected from fecal-charcoal plates using a Baermann apparatus, as previously described [63]. For brood size, survival, and hatching assays, free-living adults were briefly suspended in BU worm saline [64] in a watch glass; individual worms were then pipetted onto assay plates using a dissecting microscope (Leica S9E).

## Maintenance of *Strongyloides ratti*

*S. ratti* ED321 strain was serially passaged through female Sprague Dawley rats (Envigo) and maintained on fecal-charcoal plates as for *S. stercoralis*. Rats were inoculated with ~700 iL3s in 200 μL sterile PBS via inguinal subcutaneous injection, under isoflurane anesthesia. Collection and storage of feces containing *S. ratti* was performed as described for *S. stercoralis*.

## Maintenance of *C. elegans*

*C. elegans* N2 strain (*Caenorhabditis* Genome Center, CGC) was maintained using standard methods [65]. In brief, young adult hermaphrodites used for maintenance and thermotaxis assays were raised on 2% Nematode Growth Media (NGM) plates seeded with a lawn of *Escherichia coli* OP50 bacteria (CGC; maintenance plates). To generate age-synchronized young adults for behavioral assays, adult hermaphrodites were allowed to lay eggs on maintenance plates for 4 hours; adults were then removed, and the plates were stored at either 20˚C or 23˚C until progeny reached the desired life stage.

## Thermotaxis assays

Thermotaxis assays were performed on a custom thermoelectric behavioral arena, as previously described [49]. For some experiments, the thermoelectric control circuit included updated electrical components; a full parts list and wiring diagram are archived in a dedicated GitHub repository (https://github.com/BryantLabUW/Strongyloides-Free-Living-Thermosensory-Behaviors). Worm migrations were visualized using a 20 MP CMOS camera (Basler Ace acA5472-17um, Basler) at a frame rate of 0.2 frames per second for 45 minutes (adults) or 0.5 frames per second for 15 minutes (iL3s) using the pylon Viewer camera software suite (v7.4.0, Basler). Thermotaxis experiments were carried out on age-synchronized young adult hermaphrodites (*C. elegans*) and free-living adult males and females (*S. stercoralis*, *S. ratti*) that were incubated for at least 2 hours at the desired cultivation temperature (15˚C, 20˚C, or 23˚C). For iL3 assays, worms were collected with a Baermann apparatus and suspended in ~1 mL BU saline in a watch glass, then incubated for at least 2 hours at the appropriate temperature. For multisensory assays, 5 μL of undiluted 3-methyl-1-butanol was placed on the thermotaxis plate immediately before worms were deposited. For *post-hoc* measurements of individual worm movements, x/y coordinates were generated using the Manual Tracking plugin for Fiji, then quantified and plotted using custom MATLAB scripts (MathWorks), as previously described [55,56,66]. All custom code and hardware specifications are available on GitHub (https://github.com/BryantLabUW/Strongyloides-Free-Living-Thermosensory-Behaviors) and have been archived on Zenodo (DOI: 10.5281/zenodo.13738264).

## Brood size and survival assays

For *C. elegans*, age-synchronized adult hermaphrodites with 1–5 eggs were picked from maintenance plates onto treatment plates containing 2% NGM plates seeded with 50 μL *E. coli* HB101 bacteria (CGC) and allowed to grow for 1–3 days. For *S. stercoralis* and *S. ratti*, free-living adult females with fewer than five eggs and 3 free-living males were pipetted from a BU-filled watch glass onto treatment plates. Plates were randomly divided among the treatment temperatures (23˚C, 30˚C, or 37˚C) and moved to their respective incubators. Experiments were performed in batches of at least two temperature conditions. Worms were checked for survival (movement or pharyngeal pumping) and brood size (number of eggs and larvae on plate) every 24 hours. If the worm was still alive, it was moved to a new treatment plate and returned to its treatment incubator for another 24 hours. This process was repeated until all

worms had died. On the day each worm died, the number of eggs/larvae present when each plate was checked was recorded.

## Hatching viability assays

Age-synchronized adult hermaphrodites (*C. elegans*) and free-living adult females (*S. stercoralis*, *S. ratti*) with greater than five eggs were transferred from maintenance plates (*C. elegans*) or a BU-filled watch glass (*S. stercoralis*, *S. ratti*) onto treatment plates. All treatment plates were moved to a 23°C incubator for four hours to allow worms to lay eggs. After four hours, adult worms were removed, and the number of eggs laid on each plate was recorded ($E_0$). Plates with fewer than eight eggs were excluded from the experiment. Plates were randomly divided among the treatment temperatures (23°C, 30°C, or 37°C) and moved to their respective incubators. Experiments were done in batches of at least two temperatures at a time. After 24 hours, the number of eggs remaining on treatment plates was recorded ($E_1$). Hatching viability was then calculated using the following formula:

$$(E_0 - E_1)/E_0 * 100$$

## Statistical analysis

All statistical analyses except mean survival were conducted using GraphPad Prism 10 (Dotmatics). Mean survival values were calculate using OASIS 2 [67]. Power analyses to determine appropriate sample sizes were performed using G*Power 3.1 [68]. Statistical details for experiments are provided in figure legends and Supplemental Data File 1.

## Results

### The free-living adults of *Strongyloides* species prefer temperatures warmer than their cultivation temperature

To begin characterizing the temperature preferences of *Strongyloides* free-living adults, we first looked to see if they display behaviors similar to those of the constitutively free-living model nematode *C. elegans*. One well-documented behavior of *C. elegans* is that well-fed adult hermaphrodites migrate towards a remembered cultivation temperature when placed on a thermal gradient within their physiological temperature range (15–25°C) [34–36,39,40]. To test if *Strongyloides* free-living adults display similar thermal preferences, we generated post-parasitic free-living adults by cultivating feces containing post-parasitic larvae at 20°C for two days. Next, we placed isolated free-living females (FLFs) at 23°C in a large-format thermotaxis arena in which we established a ~21–25°C temperature gradient. Under these conditions, *C. elegans* N2 adult hermaphrodites that were raised at 20°C displayed negative thermotaxis towards their cultivation temperature, as expected (Figs 1A, 1D and S2A). In contrast, we observed that *S. stercoralis* FLFs were more likely to engage in positive thermotaxis (Figs 1B, 1D, 1E and S2A). *S. ratti* free-living females also did not migrate towards their cultivation temperature, instead displaying attraction to above-ambient temperatures, similar to *S. stercoralis* (Figs 1C–1E; and S2A). We also tested the thermal preferences of *Strongyloides* free-living males (FLMs). *S. stercoralis* FLFs and FLMs displayed similar behaviors, while *S. ratti* FLFs traveled slightly further up the gradient than *S. ratti* FLMs (S3 Fig). The difference between *S. ratti* FLFs and FLMs was not due to systemic differences in velocity, as we observed no significant difference in mean speed (mm/s) between sexes (S3C Fig).

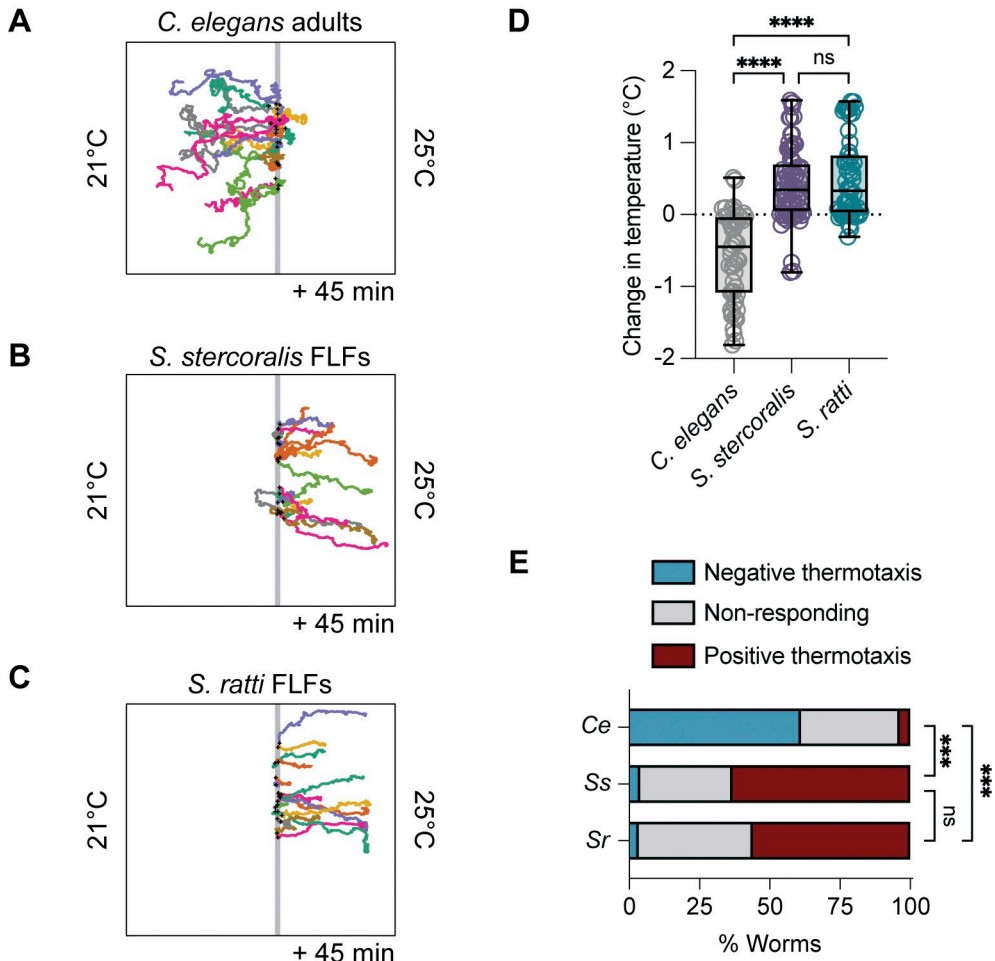

**Fig 1. Thermotaxis behaviors of *Strongyloides* free-living females near ambient temperatures are distinct from those of *C. elegans* adults.** For panels A-C, worms were placed at 23°C in a ~21–25°C gradient and allowed to migrate for 45 min. Cultivation temperature ($T_C$) = 20°C. Starting temperature ($T_{start}$) = 23°C (grey line). Black crosses show the starting positions of the worms. Randomly selected representative tracks of *C. elegans* adult hermaphrodites (A), *S. stercoralis* free-living females (FLFs) (B), and *S. ratti* FLFs (C). For all tracks, see S2 Fig. *C. elegans* hermaphrodites are seen engaging in negative thermotaxis toward their cultivation temperature, while *S. stercoralis* and *S. ratti* FLFs engage in positive thermotaxis. D) Quantification of the change in temperature experienced by *C. elegans* adults, *S. stercoralis* FLFs, and *S. ratti* FLFs. Values are the final temperature–starting temperature for each worm. Icons indicate responses of individual worms, boxes show medians and interquartile ranges, and whiskers show min and max values. n = 54 worms for *C. elegans* hermaphrodites (5 assays across 4 days), n = 76 worms for *S. stercoralis* free-living females (7 assays across 5 days), n = 59 worms for *S. ratti* FLFs (6 assays across 4 days). ns = not significant, ****$p < 0.0001$, Kruskal-Wallis test with Dunn's multiple comparisons test. E) Categorical distribution of thermotaxis behaviors in a ~21–25°C gradient across species. For each species, individual worms were considered to have engaged in positive or negative thermotaxis if their position at the end of the assay was outside of a 1 cm neutral exclusion zone centered on the starting position of each individual worm. Individuals that finished the assay within this zone were considered non-responding. n (negative/non-responding/positive) = *C. elegans*: 33/19/2; *S. stercoralis*: 3/25/48; *S. ratti*: 2/24/33. ***$p < 0.001$, Fisher's exact test with Bonferroni-Dunn correction for multiple comparisons.

## *S. stercoralis* free-living females show positive thermotaxis in conditions that elicit noxious temperature escape behavior in *C. elegans* adults

Another critical aspect of the *C. elegans* free-living thermosensory repertoire is the avoidance of noxious temperatures [37,69–71]. Experimentally, escape from noxious heat can be observed by placing adult hermaphrodites above 26°C, triggering migration down the thermal

gradient towards physiologically permissive temperatures [37,38,40]. Notably, the behavioral strategies and cellular mechanisms underlying noxious temperature responses are partially distinct from those driving innocuous temperature responses [37,50,71–73]. To determine if thermal escape behaviors are present in *Strongyloides* FLFs, we compared the migration of *C. elegans* N2 adult hermaphrodites and *Strongyloides* FLFs placed at 30˚C in a ~21–35˚C gradient (cultivation temperature, $T_C$ = 23˚C). As expected, *C. elegans* hermaphrodites showed robust migration towards cooler temperatures (Figs 2A, 2D, 2E; and S2B). Surprisingly, neither of the *Strongyloides* species displayed thermal preferences that matched those of *C. elegans* engaged in noxious temperature avoidance: *S. stercoralis* FLFs continued to show positive thermotaxis towards 34˚C and *S. ratti* FLFs did not significantly disperse from their starting temperature of 30˚C (Figs 2B–2E and S2B). The lack of dispersal by *S. ratti* FLFs is not due to reduced motility (quantified as mean speed and total distance traveled), but instead arises because their trajectories are more circuitous (S4 Fig). Finally, we tested whether *S. stercoralis* FLFs would display negative thermotaxis when exposed to mammalian core body temperatures. We found that *S. stercoralis* FLFs placed at 37˚C in a ~32–40˚C gradient did not migrate down the gradient ($T_C$ = 23˚C; S5 Fig). Together, these results indicate that the thermal preferences of *Strongyloides* free-living adults are distinct from those of *C. elegans* adult hermaphrodites, despite the similarities in their ecological niches (*i.e.*, they are both non-parasitic bacterivores). Furthermore, it suggests that the cellular mechanisms that produce robust behavioral responses to noxious heat in *C. elegans* are either not present in *Strongyloides* spp., are tuned for temperatures much higher than mammalian body heat or have been co-opted for other functions.

## Free-living adults display thermal preferences that are distinct from iL3s

The iL3s of multiple mammalian-parasitic nematode species, including *S. stercoralis* and *S. ratti*, engage in negative thermotaxis towards cooler temperatures when placed near or below their ambient cultivation temperature [55,56]. This preference for cooler temperatures may act as a mechanism to disperse iL3s into cooler soil environments where discrimination between host-emitted heat and environmental temperatures is maximized or may reflect processes to enable iL3s to disengage from migration towards non-host heat sources. What are the behaviors of *Strongyloides* FLFs in cooler gradients? We first tested the preferences of *S. stercoralis* FLFs in a ~13–23˚C gradient, conditions in which *S. stercoralis* iL3s engage in negative thermotaxis and accumulate at ~16˚C (S6A, S6E and S6F Fig) [56]. Surprisingly, we found that *S. stercoralis* FLFs displayed relatively robust positive thermotaxis in these conditions (S6B, S6E and S6F Fig). To test whether *S. ratti* display similar life-stage-specific differences in negative thermotaxis, we first tested the preferences of *S. ratti* iL3s in the ~13–23˚C gradient. We found that *S. ratti* iL3s stayed closer to their starting temperature than *S. stercoralis* iL3s (S6C, S6E and S6F). Nevertheless, *S. ratti* FLFs also engaged in positive thermotaxis when placed at 20˚C (S6D–S6F Fig). Together with our earlier experiments at above-ambient temperatures, these results indicate that *Strongyloides* FLFs are biased towards performing positive thermotaxis across a broad range of environmental temperatures.

## *Strongyloides* spp. FLFs respond to host-associated odorants in near-ambient temperature gradients

Like *C. elegans*, *Strongyloides* spp. can detect a range of non-thermosensory stimuli, including chemosensory cues [51,53,62,74]. In near-ambient temperature gradients, *S. stercoralis* iL3s prioritize thermosensory behaviors over chemosensory responses, such that exposure to temperature gradients below host body heat can override attraction to a host-associated odorant

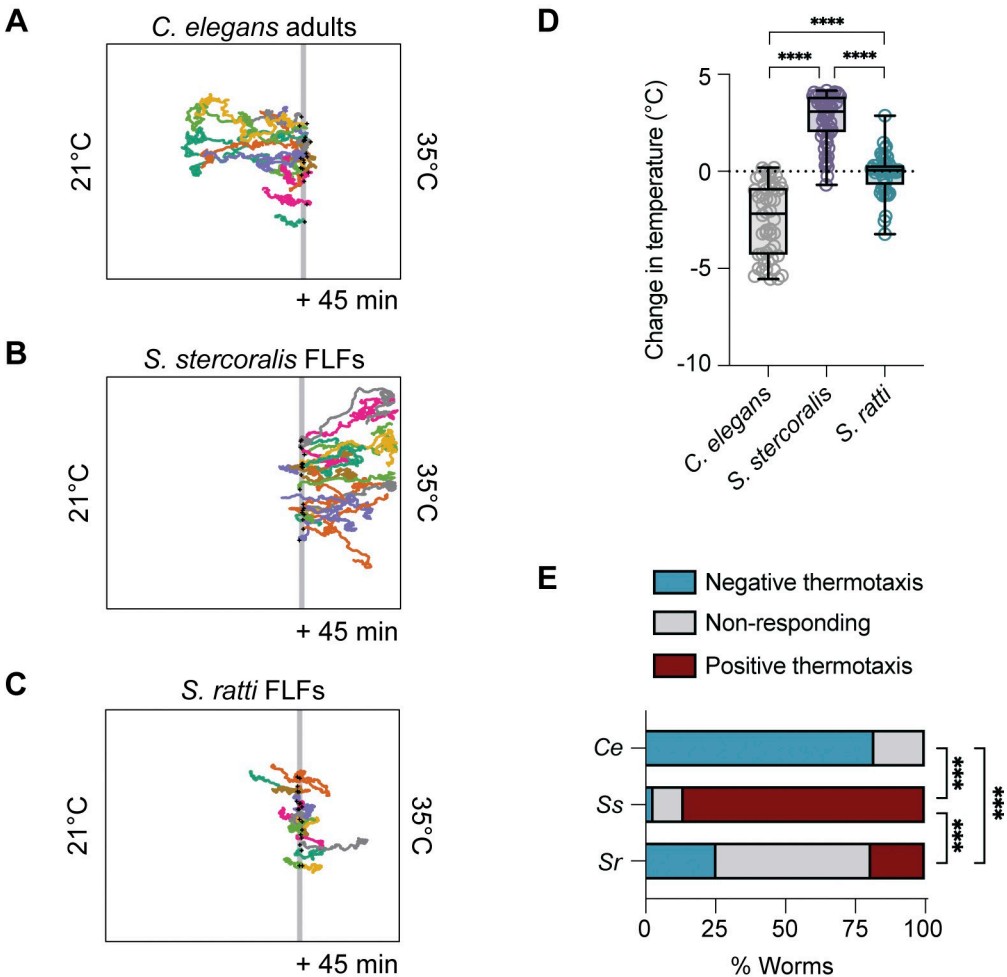

**Fig 2. *Strongyloides* free-living females do not display noxious heat avoidance near human skin temperature.** Worms were placed at 30°C in a ~21–35°C gradient and allowed to migrate for 45 min. $T_C$ = 23°C. Colored tracks represent the paths of individual worms. Grey line represents $T_{start}$ = 30°C. Black crosses show the starting positions of the worms. Randomly selected representative tracks of *C. elegans* adult hermaphrodites (A), *S. stercoralis* free-living females (B), and *S. ratti* free-living females (C). For all tracks, see S2 Fig. *C. elegans* hermaphrodites are seen engaging in noxious heat escape behaviors, while *S. stercoralis* FLFs engage in positive thermotaxis and *S. ratti* FLFs engage in neither positive nor negative thermotaxis. D) Quantification of the change in temperature for *C. elegans* adults, *S. stercoralis* FLFs, and *S. ratti* FLFs. Values are the final temperature–starting temperature for each worm. Icons indicate responses of individual worms; boxes show medians and interquartile ranges; whiskers show min and max values. n = 50 worms for *C. elegans* hermaphrodites (5 assays across 3 days), n = 65 worms for *S. stercoralis* FLFs (5 assays across 3 days), n = 47 worms for *S. ratti* FLFs (6 assays across 4 days). ****$p<0.0001$, Kruskal-Wallis test with Dunn's multiple comparisons test. E) Categorical distribution of thermotaxis behaviors in a ~21–35°C gradient across species. Individuals were considered to have engaged in positive or negative thermotaxis if their position at the end of the assay was outside of a 1 cm neutral exclusion zone centered on the starting position of each individual worm. Individuals that finished the assay within this zone were considered non-responding. n (negative/non-responding/positive) = *C. elegans*: 41/9/0; *S. stercoralis*: 2/7/56; *S. ratti*: 12/26/9. ***$p<0.001$, Fisher's exact test with Bonferroni-Dunn correction for multiple comparisons.

[49]. Do *S. stercoralis* FLFs display a similar sensory hierarchy? To test this question, we evaluated the impact of the attractive odorant 3-methyl-1-butanol (3m1b) on the behavior of *S. stercoralis* FLFs in a ~21–25°C gradient. In the absence of an odorant, *S. stercoralis* FLFs placed at 23°C displayed positive thermotaxis, traveling towards warmer temperatures (Fig 1B, 1D–1E and Fig 3). In the presence of an odorant placed either near to the starting position of

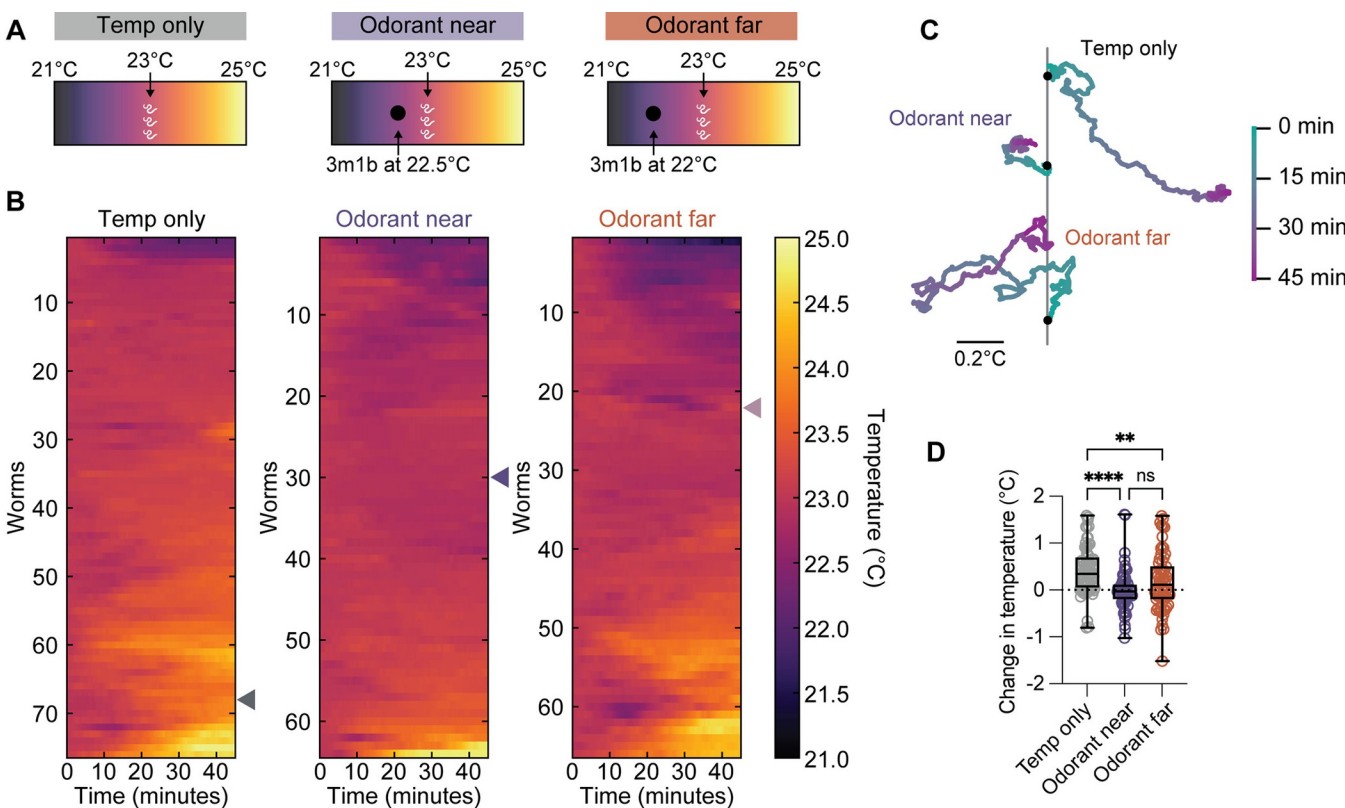

**Fig 3. *S. stercoralis* FLF's thermotaxis behavior is disrupted by the presence of an attractive odorant.** A) Diagram of the attractive odorant and temperature gradient experimental setup. Worms were placed at 23°C in a ~21–25°C temperature gradient. An attractive odorant (3m1b) was either not introduced, placed near the worm starting location (at 22.5°C), or placed far from the worm starting location (at 22°C). Assay duration: 45 minutes. Temperature-only data is reproduced from Fig 1. B) Heat map showing the temperature experienced by individual worms throughout the 45-minute assay. Cooler temperatures are represented by darker colors while warmer temperatures are represented by warmer colors. Heatmap rows represent the temperatures experienced by individual worms and are ordered by hierarchical cluster analysis such that worms with similar trajectories are grouped together. To view this data plotted as worm tracks, see S7A Fig. Triangles on the right of the heat maps indicate the individual worms chosen as representative tracks in Fig 3C. C) Example tracks of individual worms from each experimental condition. Tracks are color-coded by time in the assay as seen in the scale on the right. Gray line = 23°C ($T_{start}$); black dots indicate worm starting position. For additional representative tracks selected randomly from the full track set, see S7B Fig. D) Quantification of the change in temperature experienced by worms (final temperature–starting temperature). Icons indicate responses of individual worms, boxes show medians and interquartile ranges, and whiskers show min and max values. n = 76 worms for temperature only, n = 65 worms for odorant near (6 assays over 3 days), and n = 66 worms for odorant far (6 assays over 3 days). ns = not significant, **$p < 0.01$, ****$p < 0.0001$, Kruskal-Wallis test with Dunn's multiple comparisons test.

the worms (odorant location: 22.5°C, ~2.7 cm from $T_{start}$), or further away (odorant location: 22°C, ~5.5 cm from $T_{start}$), we observed a decrease in the distance individual worms traveled up the gradient when compared to worms in a pure thermotaxis (*i.e.*, no odorant) gradient (Figs 3 and S7). The reduction partially reflects an increase in the number of worms that initially traveled down the temperature gradient towards the odorant before traveling back up the gradient (S7 Fig). This shifted migratory pattern was particularly noticeable when the odorant was placed in the "far" location (Figs 3C and S7B). These results suggest a difference in the sensory hierarchy underlying the behaviors of *S. stercoralis* FLFs and iL3s, such that the FLF sensory hierarchy is less dominated by temperature cues (Fig 4).

## Increasing cultivation temperature decreases the lifespan of *Strongyloides* FLFs

The multifaceted repertoire of thermosensory behaviors exhibited by *C. elegans* free-living adults (*i.e.*, experience-dependent migration towards ambient cultivation temperature,

### *Strongyloides stercoralis* sensory hierarchy across life stages

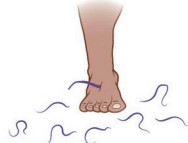

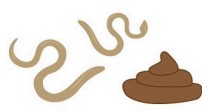

**Infective larvae**

**Free-living adults**

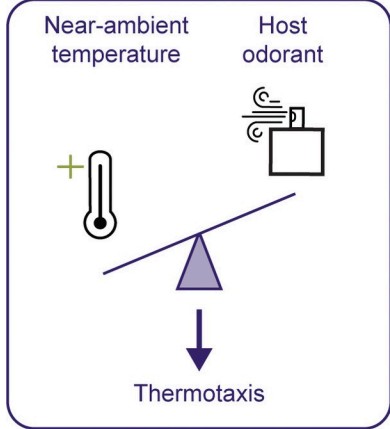

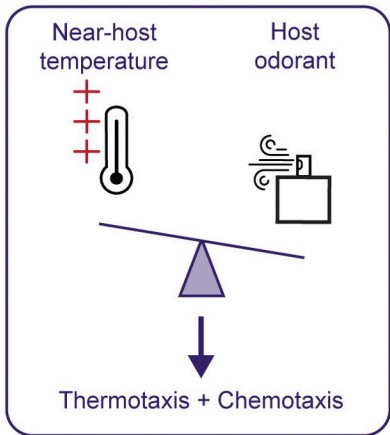

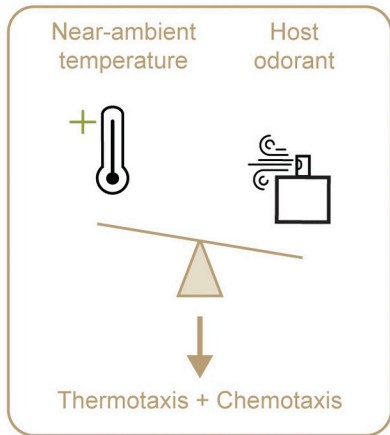

**Fig 4. Comparison of the sensory hierarchy across *Strongyloides* life stages.** Similar sensory cues result in different behaviors based on the sensory hierarchy of a specific life stage (indicated by direction of scale icon). At near-ambient temperatures, iL3s prioritize performing thermotaxis behaviors over responding to an attractive odorant [55]. At temperatures near host body heat, iL3 migration is influenced by attractive host odorants [55]. In contrast, free-living adults perform both thermotaxis and chemotaxis at near ambient temperatures, likely due to a sensory hierarchy that is less dominated by temperature cues.

isothermal tracking of cultivation temperature, and robust avoidance behaviors associated with noxious temperatures) are thought to be methods of thermoregulation in support of survival and reproduction [34–38,40,72,73]. Thus, our observation that the free-living adults of *Strongyloides* species almost exclusively engage in migration to above-ambient temperatures is highly intriguing and suggests that these warmer temperatures are potentially physiologically innocuous to the FLFs. To begin searching for an explanation for the thermophilic preferences of *Strongyloides* free-living adults, we tested the impact of different constant environmental temperatures on the survival of free-living females (Figs 5 and S8). Individual *S. stercoralis* and *S. ratti* FLFs, as well as *C. elegans* hermaphrodites, were placed on NGM plates seeded with *E. coli* HB101 at either 23˚C, 30˚C, or 37˚C and their survival was checked daily (Fig 5A). As expected, increasing the cultivation temperature of *C. elegans* adults resulted in a dramatic decrease in lifespan (Fig 5B–5E). When the cultivation temperature was increased for *S. stercoralis* and *S. ratti* FLFs, the lifespan of both *Strongyloides* species also decreased (Fig 5C–5E). Notably, this decreased survival at warmer temperatures appears to conflict with the thermal preferences of the free-living adults. This raises the question of why *Strongyloides* FLFs are attracted to temperatures that decrease their longevity.

### *S. stercoralis* FLFs show increased reproductive success at near-tropical temperatures

One possibility is that the thermal preferences of *Strongyloides* FLFs support other critical biological processes–in particular, the reproductive capacity of this life stage. To test the

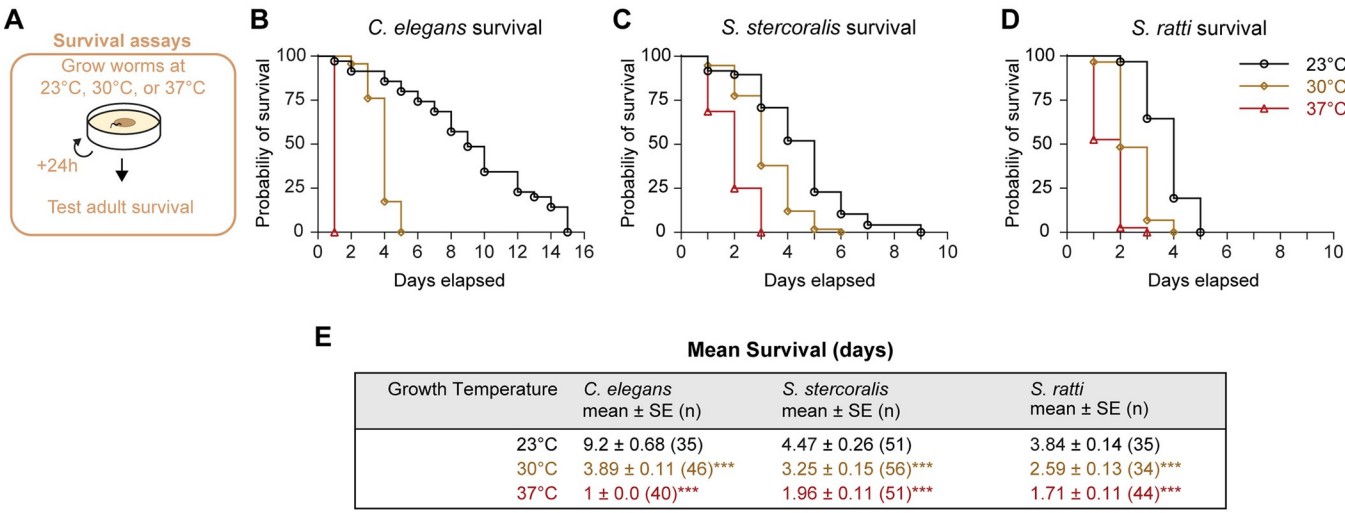

**Fig 5. Exposure to high temperatures decreases the lifespan of *Strongyloides* spp. FLFs.** A) Diagram of survival assay. Individual adults were placed on NGM plates seeded with *E. coli* HB101. The plates were then incubated at either 23˚C, 30˚C, or 37˚C. Plates were checked every 24 hours for survival. B-D) Probability of survival over time for *C. elegans* adult hermaphrodites (B), *S. stercoralis* FLFs (C), and *S. ratti* FLFs (D). Black circles represent assays run at 23˚C, gold diamonds represent assays run at 30˚C, and red triangles represent assays run at 37˚C. n = *C. elegans*: 35 (23˚C), 46 (30˚C), 40 (37˚C) worms; *S. stercoralis*: 48 (23˚C), 58 (30˚C), 48 (37˚C) worms; *S. ratti*: 31 (23˚C), 29 (30˚C), 38 (37˚C) worms. For all experiments, animals that were not found on the plate were censored. For *C. elegans* and one *S. stercoralis* experiment, the dates of censoring were not recorded; these animals (3 for *C. elegans*, 2 for *S. ratti*) have been excluded from survival analyses. Number of censored animals included in survival analyses = *C. elegans*: none; *S. stercoralis*: 5 (23˚C), 0 (30˚C), 0 (37˚C); *S. ratti*: 4 (23˚C), 5 (30˚C), 6 (37˚C). Survival curves reflect the combined survival of all worms across 5 (*C. elegans*, *S. ratti*) or 6 (*S. stercoralis*) independent experiments for each species. Survival curves for individual experiments are shown in S8 Fig. E) Table reporting average survival times for all species and temperature conditions. Values are mean ± standard error (number of animals). *p*-values are pairwise comparison of survival curves (23˚C vs 30˚C, 30˚C vs 37˚C, 23˚C vs 37˚C): *p*<0.001, Mantel-Cox test with Bonferroni-Dunn correction for multiple comparisons.

hypothesis that exposure to warmer temperatures positively impacts *Strongyloides* reproduction, we counted the number of offspring produced by individual FLFs cultivated at different temperatures (Fig 6A). Similar to previous studies, we found that exposure to warm temperatures rapidly and dramatically depresses *C. elegans* total brood size (Figs 6B and S9) [45,46]. In contrast, *S. stercoralis* FLFs showed a significant increase in total brood size at 30˚C (Fig 6B). When the temperature was increased further to 37˚C, we observed a decrease in total brood size such that the total numbers of eggs and larvae produced at 23˚C and 37˚C are not statistically different (Fig 6B). This decrease in total brood size primarily reflects the decreased lifespan of *S. stercoralis* FLFs at 37˚C. When we examined the number of eggs and larvae laid each day, we observed an increase in *S. stercoralis* daily brood size at 37˚C compared to 23˚C on the first experimental day (S9 Fig). Thus, the mechanisms that yield enhanced total brood sizes at 30˚C are likely also active at 37˚C; however, those increases are offset by decreased longevity of the FLFs. We also tested the impact of warm temperatures on *S. ratti* total brood sizes and found that they exhibited no change in total brood size between 23˚C and 30˚C and a decreased total brood size at 37˚C (Fig 6B). When we compared the *S. ratti* day-by-day brood sizes, we observed that exposure to 30˚C did drive an initial increase in daily brood size compared to 23˚C, similar to *S. stercoralis* (S9 Fig). These results suggest that *S. stercoralis* is physiologically adapted for warmer climates than *S. ratti*, which mirrors the geographical distributions of these two species: *S. ratti* is widely distributed throughout the world while *S. stercoralis* is most prevalent in tropical and subtropical climates [10,11,75].

In *C. elegans*, exposure to noxious warmth not only decreases brood size but also decreases hatching viability [48,49,76,77]. The effect of increased temperature on *Strongyloides* FLF's egg hatching viability is unknown and is particularly interesting because of our finding that

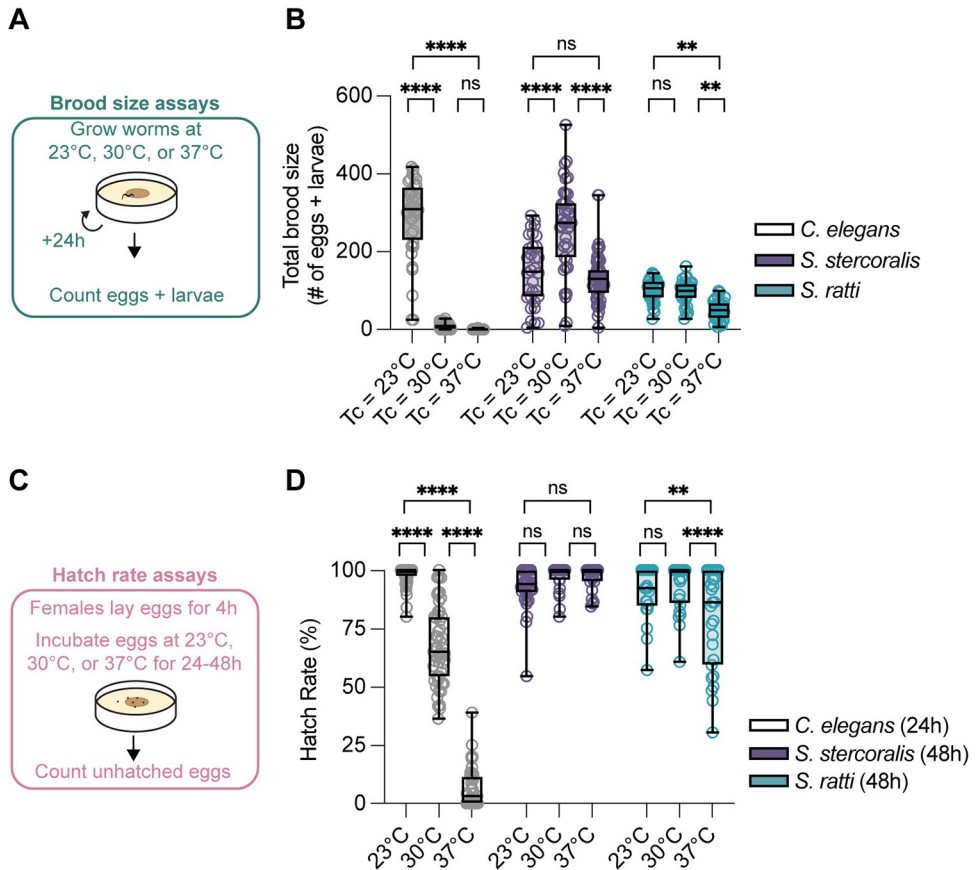

**Fig 6. Impact of near-tropical temperatures on *S. stercoralis* FLF brood size.** A) Diagram of the brood size assay. Individual adults were placed on NGM plates seeded with *E. coli* HB101. Plates were incubated at either 23°C, 30°C, or 37°C and checked every 24 hours for the number of eggs and larvae. B) Quantification of the impact of environmental temperature on brood size for *C. elegans* hermaphrodites (n = 36–47 adult worms), *S. stercoralis* free-living females (n = 32–54 adult worms), and *S. ratti* free-living females (n = 29–38 adult worms). Icons indicate responses of individual worms, boxes show medians and interquartile ranges, and whiskers show min and max values. ns = not significant, **$p<0.01$, ****$p<0.0001$, two-way ANOVA with Tukey's multiple comparisons test. C) Diagram of the hatching assay. Individual adults were placed on NGM plates seeded with *E. coli* HB101. Plates were incubated for 4 hours at 20°C. After 4 hours, females were removed from plates and eggs were counted. Plates were then incubated at 23°C, 30°C, or 37°C. After 24–48 hours, unhatched eggs were counted. D) Quantification of hatching viability for *C. elegans* (n = 50–53 plates), *S. stercoralis* (n = 23–35 plates), and *S. ratti* (n = 22–27 plates). Icons indicate responses of individual worms, boxes show medians and interquartile ranges, and whiskers show min and max values. ns = not significant, **$p<0.01$, ****$p<0.0001$, two-way ANOVA with Tukey's multiple comparisons test.

*Strongyloides* brood sizes can be enhanced at warmer temperatures. Increased brood sizes could reflect compensatory mechanisms if the hatching viability of *Strongyloides* spp. is decreased at warmer temperatures, similar to *C. elegans*. Alternatively, hatching viabilities that are largely unaffected by warmer temperatures could indicate that parasites are generally adapted to warmer environmental temperatures. To determine hatching viability, we collected synchronized eggs by letting individual *C. elegans* adult hermaphrodites and *Strongyloides* adult FLFs lay eggs onto NGM plates seeded with *E. coli* HB101 for 4 hours at 23°C. After removing the adults, we counted the number of eggs on each plate, then incubated plates at either 23°C, 30°C, or 37°C for 24 hours (*C. elegans*) or 48 hours (*Strongyloides* spp.) before re-counting any unhatched eggs (Fig 6C). As expected, we observed that *C. elegans* median hatching viability progressively decreased as a function of increased temperature (Fig 6D). In

contrast, *S. stercoralis* did not show any significant change in hatching viability as cultivation temperature increased up to 37˚C (Fig 6D). *S. ratti* also exhibited no significant change in hatching viability from 23˚C to 30˚C; however, we did observe a small, but significant, decrease in hatching viability when cultivated at 37˚C (Fig 6D). Notably, this decreased *S. ratti* hatching viability was still significantly higher than the *C. elegans* hatching viability at 37˚C ($p < 0.0001$, 2-way ANOVA with Tukey's multiple comparisons test). Together, these experiments indicate that the range of reproductively and developmentally permissive temperatures is shifted towards warmer temperatures for *Strongyloides* spp. compared to *C. elegans*.

## Discussion

Here, we present the first characterization of thermosensory behaviors by *Strongyloides* free-living adults and provide evidence to link life-stage-specific thermal preferences to physiological and reproductive consequences. We found that the free-living adults of *S. stercoralis* are attracted to above-ambient temperatures in a wide range of thermal conditions, including temperature gradients that trigger noxious heat escape behaviors in *C. elegans*, negative thermotaxis towards cultivation temperature in *C. elegans*, and negative thermotaxis towards below-ambient temperatures in *S. stercoralis* iL3s (Fig 7). We showed that unlike iL3s, *S. stercoralis* free-living females can respond to chemosensory attractants in thermal gradients below human body temperature. Impressively, we saw that some worms can even migrate against their thermal preferences in favor of moving towards the odorant. We also performed the first systematic evaluation of the impact of environmental temperatures on the physiological and reproductive potential of *S. stercoralis* free-living adults. We found that exposure to 30˚C

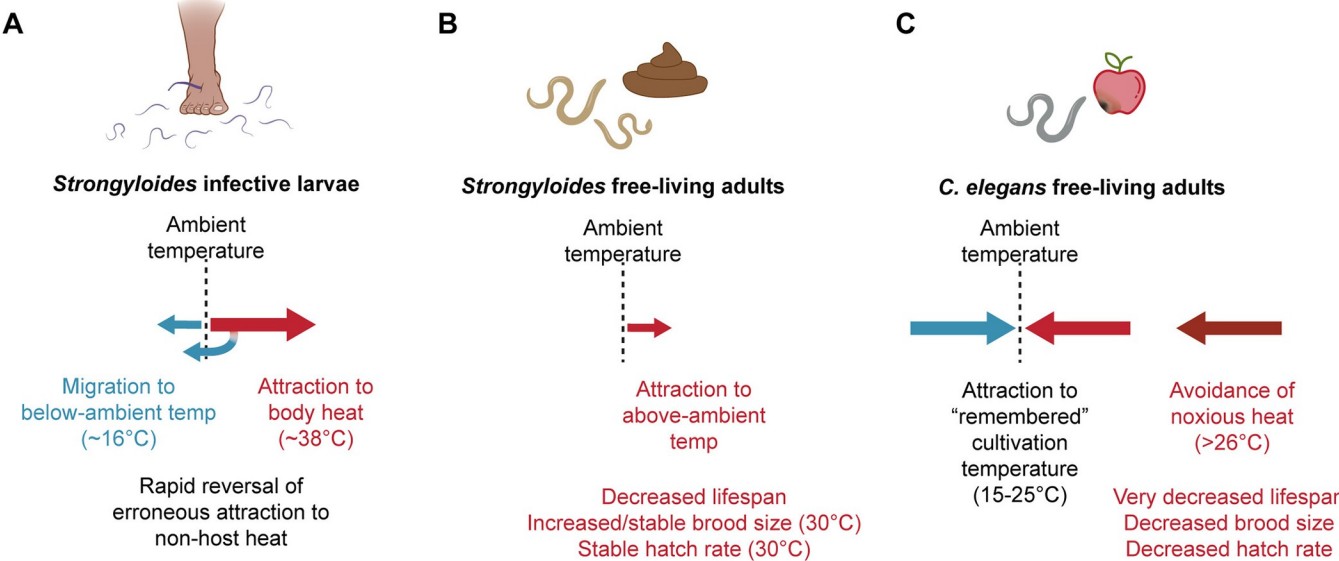

**Fig 7. Species- and life-stage-specific thermosensory behaviors and thermal physiology.** A) Summary of *Strongyloides* iL3 thermotaxis behaviors. *Strongyloides* iL3s display two modes of thermotaxis behavior: positive thermotaxis towards host body heat and negative thermotaxis towards below-ambient temperatures [55,56]. If a temperature gradient ends below ~30˚C, *Strongyloides* iL3s are able to reverse an initial attraction to warmth, performing a "U-turn" behavior that triggers sustained negative thermotaxis towards below-ambient temperatures [56]. B) Summary of *Strongyloides* FLF thermotaxis behaviors and physiological response to increased temperatures. *Strongyloides* FLFs are attracted to temperatures above ambient. Sustained exposure to above-ambient temperatures drives reductions in adult lifespan; in contrast, brood size and hatching viability are enhanced (or stable) at near-tropical temperatures. C) Summary of *C. elegans* free-living hermaphrodite thermotaxis behaviors and physiological response to increased temperatures. Between 15–25˚C, *C. elegans* adults show attraction to a "remembered" cultivation temperature and will undergo positive and negative thermotaxis towards that temperature [34–36,39,40]. At temperatures greater than 26˚C, *C. elegans* adults display a noxious heat escape response and have a very decreased lifespan, brood size, and hatching viability [34,37–39,42,43,50,69].

drives an unexpected increase in the brood size of *S. stercoralis* free-living adults, even at the expense of their longevity. Finally, comparisons between *S. stercoralis* and *S. ratti* reveal that *S. stercoralis* is adapted to warmer temperatures than *S. ratti*, a difference that aligns with their narrower geographic distribution [10,11,75].

Free-living nematodes can experience a range of environmental stressors; as ectotherms, environmental temperature is a critical source of potential stress that can significantly impact their individual fitness. The free-living model nematode *C. elegans* displays a complex thermoregulatory process that, in response to thermal stress, prioritizes individual survival and future fecundity over immediate production of offspring [78,79]. Major elements of the *C. elegans* thermoregulatory response include thermotaxis navigation towards physiologically permissive temperatures, inhibition of reproduction, and altering a range of other physiological processes [34,43,45,48,78,79]. Until now, it has been unclear whether this thermoregulatory strategy was a ubiquitous feature of ectothermic, bacterivorous nematodes. Our results demonstrate that the free-living adults of two nematode species, *S. stercoralis* and *S. ratti*, display an alternative thermoregulatory strategy, one that prioritizes immediate expansion of infective larval populations over individual adult survival. These findings demonstrate that despite their similarities to *C. elegans*, *Strongyloides* free-living adults display distinct thermosensory responses and physiological characteristics. We note that we used the *C. elegans* N2 strain, which has likely adapted to maintenance in laboratory environments. Previous studies have found extensive variation in the lifetime fecundity and temperature preferences of geographically distinct *C. elegans* isolates, indicating that thermoregulatory strategies can vary even within a single species [46,48,80,81]. Furthermore, our experiments utilize 2D linear environments and stable environmental temperatures; exposure to more complex 3D environments that display diurnal fluctuations in temperatures could reveal more nuanced behaviors across both *C. elegans* and *Strongyloides* species.

Previous studies have found large differences in thermal preferences between *C. elegans* adults and host-seeking *Strongyloides* iL3s [55,56]. Our current results expand these differences to *Strongyloides* free-living adults. The specialized thermotaxis behaviors of *S. stercoralis* iL3s are thought to arise from adaptations to a neural circuit that is conserved with *C. elegans*. Our findings suggest that these adaptations are a persistent feature of the *Strongyloides* thermosensory circuit across life stages. However, our data also reveals plasticity in the thermosensory behaviors of *Strongyloides* as a function of life stage, specifically, the failure of *Strongyloides* FLFs to perform negative thermotaxis, a behavior that is robustly present in *Strongyloides* iL3s. These differences mirror the life-stage-specific differences in chemosensory preferences previously observed with *Strongyloides* species [52,53,62]. In *C. elegans*, changes in the behavioral valence of sensory stimuli across life stages can arise from plasticity in the functional architecture of sensory neural microcircuits [72,82,83]. Similar life-stage-specific changes in functional connectivity could account for the behavioral differences we see between *Strongyloides* iL3s and free-living adults. Taken together, our results emphasize the need for future studies that pinpoint the neural mechanisms that drive parasite-specific and life-stage-specific thermotaxis behaviors in *Strongyloides* species.

Our experiments demonstrate that soil-resident *Strongyloides* free-living adults cannot survive prolonged exposure to temperatures approximating mammalian body heat. This physiological inability contrasts strongly with the ability of host-resident *Strongyloides* parasitic adults to survive for many months within host bodies, as well as the ability of soil-resident iL3s to survive the thermal transition associated with host invasion [84]. Thus, ectothermic species that parasitize warm-blooded animals are not innately resistant to the negative impacts of thermal stress. What are the molecular and cellular mechanisms that not only enable *Strongyloides* parasitic adults to survive and reproduce within the host environment, but also produce a

remarkable increase in maximum lifespan compared to free-living adults [30,84,85]? If these mechanisms are actively acquired or maintained following host infection, they may represent an exciting source of molecular targets for novel anthelmintic drugs. In the future, we hope that research leveraging our mechanistic understanding of *C. elegans* thermal physiology and the growing *Strongyloides* genetic toolkit will provide insight into the remarkable physiological abilities of parasitic adults [74,86].

Here, we have focused on understanding the thermotaxis behaviors and thermal physiology of *Strongyloides* free-living adults, a life stage observed only in *Strongyloides* and the closely related *Parastrongyloides* genera [87]. Although other medically important parasitic species, such as human hookworms, lack a free-living generation, they do deposit eggs and larvae into the environment that must also survive environmental temperatures long enough to mature into infective larvae that will locate and infect host animals. Is the ability of *Strongyloides* post-parasitic eggs to maintain high hatching viability in warm temperatures a common feature of nematodes that are endemic to tropical and sub-tropical regions? Tropical isolates of the free-living nematode *Caenorhabditis briggsae* are more cryophilic than even *C. elegans*, suggesting that preference for warm temperatures is not an intrinsic feature of nematodes found in tropical climates [49,88]. For parasitic nematodes, recent experimental evidence for species adaptation to warm temperatures is limited and potentially determined by experimental conditions: whereas one study reports that the eggs of *Necator americanus*, a human hookworm, display low rates of hatching at temperatures above 35˚C, another reports that the eggs of both *N. americanus* and the human hookworm *Ancylostoma duodenale* are able to hatch (albeit with increasing mortality) at temperatures up to 40˚C [89,90]. The eggs of other parasitic nematode species, including the pig large roundworm *Ascaris suum*, also display remarkable resistance to elevated temperatures [91–95]. For behavioral preferences, previous studies have found that *S. stercoralis* iL3s prefer temperatures warmer than the human hookworm *Ancylostoma ceylanicum* and other mammalian-parasitic nematodes [55]. Thus, the ability of *S. stercoralis* FLFs to maintain high hatching viability at 37˚C may indicate a particularly high thermal tolerance, even among human-parasitic nematodes. Nevertheless, mammalian parasites must all survive and reproduce at body heat, suggesting the possibility of common molecular mechanisms for avoiding decreased longevity and diminished fecundity due to heat stress.

Our investigation of the impact of environmental temperatures on the effective fecundity of the *Strongyloides* free-living generation has intriguing implications for the role of this life stage in promoting infections across diverse climates. Our experiments show that the reproductive potential of *Strongyloides* free-living adults is relatively impervious to prolonged exposure to 30˚C. Human infections with *S. stercoralis* predominantly occur in tropical and sub-tropical climates [10,11]. Our results suggest that the specialized physiology of *S. stercoralis* can contribute to the maintenance of infectious populations within these climates. Specifically, our findings deepen our understanding of the complex relationship between temperature, the choice between the heterogonic and homogonic developmental pathways, and the production of infective larvae. Previous work has shown that *S. stercoralis* post-parasitic L1 larvae that experience temperatures below 34˚C are driven to develop into free-living adults [21]. Now, we show that after reaching adulthood, *S. stercoralis* adults are attracted to warm temperatures, potentially enhancing their reproductive output at the expense of individual longevity. Furthermore, even in cases where total reproductive output is only maintained, rather than enhanced (*e.g.*, *S. stercoralis* at 37˚C), exposure to warmer temperatures will shorten the time required to generate large numbers of iL3s, which may influence the functional infectivity of the soil-dwelling population [96–98].

Together, our results suggest that the thermotaxis behaviors and thermal physiology of *S. stercoralis* are particularly adapted for tropical climates. These findings are especially

concerning in the context of anthropogenic climate change, where increases in global temperatures could shift the geographic ranges capable of supporting *S. stercoralis* transmission while also improving reproductive capacity and functional infectivity of these parasites [99,100]. Ultimately, our work emphasizes the importance of developing a deeper understanding of the complex behaviors and physiology of *S. stercoralis*, a highly neglected source of human disease.

## Supporting information

**S1 Fig. Life cycles of *S. stercoralis* and *S. ratti*.** A) The life cycle of *S. stercoralis* in a human host [8]. 1) Infection starts when soil-dwelling infective third-stage larvae (iL3s) locate a host and skin penetrate. 2) Inside the host, iL3s migrate to the small intestine and develop into reproductively active parasitic females. The eggs and larvae of parasitic females are voided from the host in feces, where they can either develop directly into iL3s (3a) or into non-parasitic "free-living" adults that engage in sexual reproduction (3b). The offspring of the free-living adults all develop into iL3s. In *S. stercoralis*, autoinfection can occur when eggs hatch within the host and develop into autoinfective larvae. B) The life cycle of *S. ratti* in a rat host [19]. The stages of infection are similar to *S. stercoralis*, with the following differences: a) the progeny of parasitic females leave the host exclusively as unhatched eggs; b) the lack of precocious hatching precludes autoinfection of original hosts.
(TIF)

**S2 Fig. Expanded plots of thermotaxis trajectories across species and thermal gradients.** All tracks of *C. elegans* adult hermaphrodites, *S. stercoralis* free-living females (FLFs), and *S. ratti* FLFs migrating in either a ~21–25°C gradient (A) or a ~21–35°C gradient (B). For panel A, cultivation temperature ($T_C$) = 20°C and starting temperature ($T_{start}$) = 23°C (grey line). A subset of the tracks plotted here is also shown in Fig 1. For panel B, $T_C$ = 23°C and starting temperature ($T_{start}$) = 30°C (grey line). A subset of the tracks plotted here is also shown in Fig 2. For the ~21–25°C gradient, n = 54 worms for *C. elegans* hermaphrodites (5 assays across 4 days), n = 76 worms for *S. stercoralis* free-living females (7 assays across 5 days), n = 59 worms for *S. ratti* FLFs (6 assays across 4 days). For the ~21–35°C gradient, n = 50 worms for *C. elegans* hermaphrodites (5 assays across 3 days), n = 65 worms for *S. stercoralis* FLFs (5 assays across 3 days), n = 47 worms for *S. ratti* FLFs (6 assays across 4 days).
(TIF)

**S3 Fig. *Strongyloides* free-living males display thermal preferences similar to free-living females.** A) Individual tracks of *S. stercoralis* free-living males responding to a ~21–25°C temperature gradient ($T_{start}$ = 23°C). $T_C$ = 20°C. Each colored line is the track of an individual *S. stercoralis* male's path throughout the 45-minute assay. Black crosses represent the starting positions of each worm. The grey line represents $T_{start}$ = 23°C. B) Individual tracks of *S. ratti* free-living males responding to a temperature gradient from ~21°C to 25°C when placed at 23°C. Each colored line is the track of an individual *S. stercoralis* male's path throughout the 45-minute assay. Black crosses represent the starting positions of each worm. The grey bar represents a 1°C-wide starting zone of the assay and is centered on 23°C. C) Quantification of the change in temperature (left) and mean speed (right) for free-living males (FLMs) vs. free-living females (FLFs) of *S. stercoralis* (purple) or *S. ratti* (teal). Icons indicate responses of individual worms, boxes show medians and interquartile ranges, and whiskers show min and max values. *S. stercoralis* FLMs and FLFs migrated similarly in the temperature gradient; *S. ratti* FLMs showed reduced positive thermotaxis relative to *S. ratti* FLFs. The mean speed of worms in the gradient was not significant between the sexes. n = 76 worms for *S. stercoralis* FLFs (7 assays over 5 days), n = 49 worms for *S. stercoralis* FLMs (5 assays over 3 days), n = 59 for *S. ratti*

FLFs (6 assays over 4 days), and n = 49 for *S. ratti* FLMs (5 assays over 4 days). Icons indicate responses of individual worms, boxes show medians and interquartile ranges, and whiskers show min and max values. ns = not significant, \*\*\*$p<0.001$, two-way ANOVA with Šídák's multiple comparisons test. D) Categorical distribution of thermotaxis behaviors in a ~21–25˚C gradient across species. Individuals were considered to have engaged in positive or negative thermotaxis if their position at the end of the assay was outside of a 1 cm neutral exclusion zone centered on the starting position of each individual worm. Individuals that finished the assay within this zone were considered non-responding. n (negative/non-responding/positive) = *S. stercoralis* FLF: 3/25/48; *S. ratti* FLF: 2/24/33; *S. stercoralis* FLM: 0/19/30; *S. ratti* FLM: 6/27/16. ns = not significant, Fisher's exact test with Bonferroni-Dunn correction for multiple comparisons.
(TIF)

**S4 Fig. Expanded quantifications of worm mobility across species and thermal gradients.** Quantification of the total distance traveled (A), mean speed (B), and distance ratio (C) of worms placed in either the ~21–25˚C gradient (teal) or the ~21–35˚C gradient (rose). Distance traveled was calculated for each worm as the summation of Euclidean distances between sequential measured X/Y coordinates. Distance ratio was calculated for each worm by dividing the total distance traveled by the maximum Euclidean displacement from its starting position. A larger distance ratio represents a more circuitous path; a distance ratio of 1 is a completely straight path. For all panels, icons indicate responses of individual worms, boxes show medians and interquartile ranges, and whiskers show min and max values. ns = not significant, \*\*\*$p = 0.0001$, \*\*\*\*$p<0.0001$, two-way ANOVA with Sidak's multiple comparisons test. For the ~21–25˚C gradient, n = 54 worms for *C. elegans* hermaphrodites (5 assays across 4 days), n = 76 worms for *S. stercoralis* free-living females (7 assays across 5 days), n = 59 worms for *S. ratti* FLFs (6 assays across 4 days). For the ~21–25˚C gradient, n = 50 worms for *C. elegans* hermaphrodites (5 assays across 3 days), n = 65 worms for *S. stercoralis* FLFs (5 assays across 3 days), n = 47 worms for *S. ratti* FLFs (6 assays across 4 days).
(TIF)

**S5 Fig. *S. stercoralis* FLFs fail to display noxious heat escape behaviors at host body temperatures.** A) Individual tracks of *S. stercoralis* free-living females responding to a ~32–40˚C temperature gradient ($T_{start}$ = 38˚C). $T_C$ = 23˚C. Each colored line is the trajectory of an individual worm during the 45-minute assay. Black crosses represent the starting positions of each worm. The grey line represents $T_{start}$ = 38˚C. B) Quantification of the change in temperature for *S. stercoralis* free-living females. Values are final temperature–starting temperature for each worm. Icons indicate responses of individual worms; boxes show medians and interquartile ranges; whiskers show min and max values. ns = not significantly different from a hypothetical value of 0, Wilcoxon signed-rank test. C) Categorical distribution of *S. stercoralis* thermotaxis behavior in a ~32–40˚C gradient. Individuals were considered to have engaged in positive or negative thermotaxis if their position at the end of the assay was outside of a 1 cm neutral exclusion zone centered on the starting position of each individual worm. Individuals that finished the assay within this zone were considered non-responding. n (negative/non-responding/positive) = 9/14/7.
(TIF)

**S6 Fig. Conditions that elicit negative thermotaxis in *Strongyloides* iL3s do not elicit negative thermotaxis in *Strongyloides* FLFs.** A-D) Tracks of *S. stercoralis* iL3s (A), *S. stercoralis* FLFs (B), *S. ratti* iL3s (C), and *S. ratti* FLFs (D) in a ~13–23˚C gradient ($T_{start}$ = 20˚C, $T_C$ = 23˚C). Each colored line is an individual worm's path throughout the 15-minute assay. Black crosses represent the starting positions of each worm. The grey line represents $T_{start}$ = 20˚C.

Assay duration: 45 minutes (adults) or 15 minutes (iL3s). E) Quantification of the change in temperature experienced by worms in the ~13–23˚C gradient. Icons indicate responses of individual worms, boxes show medians and interquartile ranges, and whiskers show min and max values. n = 30–50 worms from 3–5 assays. ***$p<0.001$, ****$p<0.0001$, two-way ANOVA with Tukey's multiple comparisons test. F) Categorical distribution of thermotaxis behaviors in a ~13˚C-23˚C gradient across species. Individuals were considered to have engaged in positive or negative thermotaxis if their position at the end of the assay was above or below 0.5˚C of their starting position, respectively. Individuals that finished the assay within 0.5˚C above or below $T_{start}$ were considered non-responding. ***$p<0.001$, Fisher's exact test with Bonferroni-Dunn correction for multiple tests.
(TIF)

**S7 Fig. Expanded quantifications of the impact of an attractive odorant on the thermotaxis behaviors of *S. stercoralis* FLFs.** A) Tracks of worms in a ~21–25˚C pure thermotaxis gradient or a thermal gradient with an attractive odorant (3m1b, pure) placed near (at 22.5˚C) or far (at 22˚C) from $T_{start}$ (23˚C). Assay duration: 45 minutes. Colored tracks represent the path of individual worms. Black crosses represent the starting location of individual worms. The grey line represents $T_{start}$ = 23˚C. B) Randomly selected representative tracks of individual *S. stercoralis* FLFs in each experimental condition from panel A. Tracks are color-coded by time. Vertical red hash lines represent the position of the odorant. Black crosses show the starting location of individual worms. C) Quantification of the minimum temperature experienced by *S. stercoralis* FLFs in a ~21–25˚C gradient with and without 3m1b. The minimum temperature experienced in the presence of an odorant was significantly lower than in the temperature-only condition. Icons indicate responses of individual worms, boxes show medians and interquartile ranges, and whiskers show min and max values. ns = not significant, ***$p<0.001$, ****$p<0.0001$, Kruskal-Wallis test with Dunn's multiple comparisons test. D) Quantification of the maximum temperature experienced by *S. stercoralis* FLFs in a ~21–25˚C gradient with and without 3m1b. The maximum temperature experienced in the presence of an odorant was significantly lower than in the temperature-only condition. n = 76 worms for temperature only (7 assays over 5 days), n = 65 worms for odorant near (6 assays over 3 days), and n = 66 worms for odorant far (6 assays over 3 days). *$p<0.05$, ****$p<0.0001$, Kruskal-Wallis test with Dunn's multiple comparisons test. E) Quantification of the starting temperature experienced by *S. stercoralis* FLFs in a ~21–25˚C gradient with and without 3m1b. The starting temperatures of worms were not significantly different between the experimental conditions. ns = not significant, Kruskal-Wallis test with Dunn's multiple comparisons test.
(TIF)

**S8 Fig. Kaplan-Meier survival curves for individual experiments.** Probability of survival over time for *C. elegans* adult hermaphrodites, *S. stercoralis* FLFs, and *S. ratti* FLFs. Plots show survival curves for independent experiments. n = number of events (number of censored worms). When the number of censored worms is marked with an asterisk (*), the date of censoring is unknown, and the censored worms are excluded from plots and statistical analyses.
(TIF)

**S9 Fig. Impact of environmental temperature on daily brood sizes across species.** A-C) Brood size (eggs + larvae) recorded each day for *C. elegans* adults (A), *S. stercoralis* FLFs (B), and *S. ratti* FLFs (C) as a function of incubation temperature. Grey circles = 23˚C; yellow diamonds = 30˚C; red triangles = 37˚C. Icons indicate brood sizes of individual worms, boxes show medians and interquartile ranges, and whiskers show min and max values. D) Day one brood size across species. *S. stercoralis* FLFs showed a significant increase in brood size on day

1 of the brood size assay at 30˚C and 37˚C. *S. ratti* FLFs showed a significant increase in brood size on day 1 of the brood size assay at only 30˚C. In contrast, *C. elegans* adults showed a significant decrease in brood size on day 1. n = 31–54 adult worms. ns = not significant, ****$p<0.0001$, two-way ANOVA with Tukey's multiple comparisons test.
(TIF)

**S1 File. This file includes all data used for statistical analyses and the results of statistical tests, including exact *p* values.**
(XLSX)

## Acknowledgments

We gratefully acknowledge Neil E. Warren and the University of Washington Instrumentation Services. We thank Dr. Dana Miller and the Hot Buttered Mice group for their help regarding survival assays and OASIS 2. We thank Yi Zhang, Kyle Thieringer, Rachel Oaks-Leaf, and Dr. Dana Miller for insightful comments on the manuscript.

## Author Contributions

**Conceptualization:** Elissa A. Hallem, Astra S. Bryant.

**Data curation:** Ben T. Gregory, Astra S. Bryant.

**Formal analysis:** Ben T. Gregory, Mariam Desouky, Jaidyn Slaughter, Astra S. Bryant.

**Funding acquisition:** Elissa A. Hallem, Astra S. Bryant.

**Investigation:** Ben T. Gregory, Mariam Desouky, Jaidyn Slaughter, Astra S. Bryant.

**Methodology:** Astra S. Bryant.

**Project administration:** Astra S. Bryant.

**Software:** Astra S. Bryant.

**Supervision:** Astra S. Bryant.

**Validation:** Astra S. Bryant.

**Visualization:** Ben T. Gregory, Astra S. Bryant.

**Writing – original draft:** Ben T. Gregory, Astra S. Bryant.

**Writing – review & editing:** Ben T. Gregory, Elissa A. Hallem, Astra S. Bryant.

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
