## [Decision Letter · Decision Letter 0]

18 Oct 2024

Dear Dr Bryant,

Thank you very much for submitting your manuscript "Thermosensory behaviors of the free-living life stages of *Strongyloides*species support parasitism in tropical environments" for consideration at PLOS Neglected Tropical Diseases. As with all papers reviewed by the journal, your manuscript was reviewed by members of the editorial board and by several independent reviewers. The reviewers appreciated the attention to an important topic. Based on the reviews, we are likely to accept this manuscript for publication, providing that you modify the manuscript according to the review recommendations. 

Sincerely,

Eduardo José Lopes-Torres, Ph.D.

Academic Editor

Jong-Yil Chai

Section Editor

Reviewer's Responses to Questions

**Key Review Criteria Required for Acceptance?**

**Methods**

-Are the objectives of the study clearly articulated with a clear testable hypothesis stated?

-Is the study design appropriate to address the stated objectives?

-Is the population clearly described and appropriate for the hypothesis being tested?

-Is the sample size sufficient to ensure adequate power to address the hypothesis being tested?

-Were correct statistical analysis used to support conclusions?

-Are there concerns about ethical or regulatory requirements being met?

Reviewer #1: (No Response)

Reviewer #2: All appropriate I have no specific comments here

Reviewer #3: The objectives of the study are clearly articulated, the experimental design was appropriate.

I am not aware of ethical concerns here.

**Results**

-Does the analysis presented match the analysis plan?

-Are the results clearly and completely presented?

-Are the figures (Tables, Images) of sufficient quality for clarity?

Reviewer #1: (No Response)

Reviewer #2: Results are exceptionally clearly presented in an easily digestible form. Notably the presentation of the data in figure format is excellent. The use of figures to illustrate the concepts conveyed is also particularly helpful.

Reviewer #3: The results are clearly presented and in line with the objectives of the study. The Figures are very nicely layed out.

**Conclusions**

-Are the conclusions supported by the data presented?

-Are the limitations of analysis clearly described?

-Do the authors discuss how these data can be helpful to advance our understanding of the topic under study?

-Is public health relevance addressed?

Reviewer #1: (No Response)

Reviewer #2: All supported fully by data and consider the future direction of the work adequately.

Reviewer #3: The authors' conclusions are supported by the data. Public health relevance is addressed.

**Editorial and Data Presentation Modifications?**

Reviewer #1: (No Response)

Reviewer #2: No issues

Reviewer #3: I recommend accepting this manuscript with only minor changes:

- a clarification/discussion of the existence of variable thermal preference among nematodes of the Caenorhabditis gender and even within C. elegans species. C. elegans was used as a 'non-parasitic' nematode control, which makes sense because it is the best studied nematode in term of thermosensory behavior. But it remains one particular case. Disscussing what's known a little more largely would be interesting (no new data needed). See below.

**Summary and General Comments**

Reviewer #1: In this manuscript the authors compare the thermotactic behavior of the facultative free-living adults of two species of the nematode genus Strongyloides, which are small intestinal parasites and compare it with the corresponding behavior of the infective larvae of the same species and of adults of the non-parasitic model nematode C. elegans. The study is interesting and well done. The manuscript is very well written and easy to follow. The rational and the results are presented in a well-understandable form. Below, I list a number of points, I recommend the authors to address prior to acceptance of this manuscript. Only one of these points is major.

Specific comments

Major point: 

Although the authors do mention in the introduction that there are biological within species differences between different isolates, they interpret their results in this study always with respect to species. Given that for each species only one isolate (the respective laboratory isolate) was analyzed this results in an overinterpretation of the data. I do not think that it is clear if the differences observed between S. stercoralis and S. ratti are differences between species or between isolates. I am not asking for the inclusion of additional isolates but the authors should discuss more clearly that their data are valid for the particular isolates and that it is not known if similar differences might also exist between isolates of the same species.

Minor points:

Fig. 1E (similar in Fig. S2D): It is indicated that there are a few Strongyloides with negative thermotaxis but in Fig. 1BC no such examples are shown. Although the authors do mention that they show representative (and not all) tracks, this is a bit confusing.

Line 289: At this place it is not clear if the temperatures tested are indeed noxious for Strongyloiodes.

Line 291: Please comment if the S. stercoralis were still fully motile at 30°C.

Lines 309,313: It is a bit confusing that in the text the gradient is described as 12-22°C but in Fig. S5 it is labelled as 13-23°C. Similar small inconsistencies occur also at other places, for example between Fig. S6A and its legend.

Fig. S4 is not mentioned in the text

Line 315: "Fig. S5D, F" (the 5 is missing).

Fig 3B: I cannot really understand how to interpret the heatmap. 

Fig. 4: Is this figure really necessary?

Lines 619,620 (legend to Fig. 5): Please indicate here how many worms per experiment.

Lines 675,676: Mention that autoinfection is specific for S. stercoralis and does not occur in S. ratti.

Reviewer #2: Gregory and colleagues present an elegant study of the thermal preferences and of Strongyloides free-living adult nematodes in comparison to the well-studied preferences of C. elegans. This work builds upon previous studies from the authors on the thermal and odorant preferences of infective-stage Strongyloides. The data presented demonstrate that Strongyloides FL adults are attracted to higher temperatures but that at these temperatures' lifespan is reduced. Interestingly at higher temperatures Strongyloides FL adults appear to compensate for reduced lifespan by increasing reproductive capacity; this is especially true for S. stercoralis which is more adapted to tropical environments in comparison to S. ratti which has a global distribution. 

This manuscript not only provides comparative data for nematodes which occupy similar environmental niches (C. elegans and Strongyloides) but also begins to provide insight into the stage-specific behaviours of parasitic nematodes which facilitate their success. It also presents a novel dataset on which to build functional studies that can tease apart the underlying mechanisms responsible for such behaviours. 

The broader impact of these findings on the understanding of parasite transmission and infection in the context of climate change and, the potential to leverage such knowledge in pursuit of novel forms of parasite control, is particularly significant. This is an exceptionally well written manuscript built upon well considered hypothesis and experiments and illustrated with informative figures that are presented beautifully. As such, I have no significant suggestions for improvement and congratulate the authors on their work. I am keen to see this study accepted for publication in its current form.

Reviewer #3: Soil-transmitted parasitic nematodes represent a major global health threat, with Strongyloides stercoralis being a potentially fatal parasite common in tropical regions. Unlike other human-parasitic nematodes, Strongyloides has a unique life cycle that includes a free-living generation where all offspring become infective larvae. Previous studies characterized the thermotactic behavior of infective forms of Strongyloides, as well as the interplay with chemotaxis to form an infection-promoting strategy. In contrast, the sensory behaviors that allow the free-living adult forms to navigate soil environments was unknown. In the present study, the behavioral thermal preference, temperature-dependent survival and reproductive potential, as well as the interplay with chemotaxis was characterized in S. stercoralis, S. ratti (a rat infecting relative species) and C. elegans (a non-parasitic free-living nematode with well characterized thermo-sensory behavior, used for comparison purpose). The frere-living forms of the two parasitic specie were found to strongly diverge frorm that of C. elegans, with a constitutive positive thermotaxis. Higher temperatures shorten the lifespan of Strongyloides, like in C. elegans. However, unlike in C. elegans, higher temperature enhanced the reproductive potential in S. stercoralis. Moreover, chemotaxis prevailed over thermotaxis in the parasitic species free living forms, which is different to the situation in infective larvae. Together, these results suggest that “living form-specific” thermotaxis and chemotaxis modulation within the same species might constitute a potential strategy to optimize parasite transmission and may explain Strongyloides' prevalence in tropical climates.

The study is very well written, results presented in efficient and beautiful figures and the overall conclusions are supported by the data. The results are of potential relevance for tropical disease understanding and certainly deserve being published as they are right now. I congratulate the author for this very interesting study. I only have minor concerns, not requiring additional experiments:

1) I understand the choice of C. elegans as comparison point as it is very well studied. The classical N2 strain used here was originally isolated in Bristol, UK, and potentially further drifted genetically in the lab before being fixed as reference strain. It would have been interesting to compare Strongyloides with a free-living tropical species (e.g, the relatively well-studied C. briggsae). Obviously, it is too late to change this. But the authors could at least discuss this point and make it more apparent that the C. elegans strain used as ‘reference’ free-living species also diverges quite a lot in term of habitat. 

2) Along the same line, it would be fair to mention a previous study in C. briggsae doi:10.1242/jeb.075408 . My understanding of this article is that tropical isolate of this species were performing robust negative thermotaxis. For example, JU1341 was isolated in Kerala, India, and behaved very similarly to N2. What’s more several tropical isolates displayed even more pronounced cryophilic drives, which is opposite to free-living Strongyloides in the present study, and which reinforces the idea that the positive thermotaxis (+ physiological adaptation to higher temperature for growth) in the parasitic species is really part of a move to favor future infectivity in the next generation.

3) Another limitation is that we don’t know to what extent the linear gradient assay in 2D will resemble what the worms are doing in the soil (complex 3D environment). Even if this limitation regards virtually every studied conducted with nematode in lab settings so far, it would be fair to the reader to brought it up explicitly.

PLOS authors have the option to publish the peer review history of their article (what does this mean?). If published, this will include your full peer review and any attached files.

Reviewer #1: No

Reviewer #2: No

Reviewer #3: No

Figure Files:

Data Requirements:

Reproducibility:

References

---

## [Editor Report · Decision Letter 1]

27 Nov 2024

PNTD-D-24-01299R1Thermosensory behaviors of the free-living life stages of *Strongyloides* species support parasitism in tropical environmentsPLOS Neglected Tropical Diseases

Dear Dr. Bryant, Thank you for submitting your manuscript to PLOS Neglected Tropical Diseases. After careful consideration, we feel that it has merit but does not fully meet PLOS Neglected Tropical Diseases's publication criteria as it currently stands. Therefore, we invite you to submit a revised version of the manuscript that addresses the points raised during the review process. Please submit your revised manuscript within 30 days Dec 27 2024 11:59PM. If you will need more time than this to complete your revisions, please reply to this message or contact the journal office at plosntds@plos.org. Please include the following items when submitting your revised manuscript:

*
A marked-up copy of your manuscript that highlights changes made to the original version. You should upload this as a separate file labeled 'Revised Manuscript with Track Changes'. *
An unmarked version of your revised paper without tracked changes. You should upload this as a separate file labeled 'Manuscript'.

We look forward to receiving your revised manuscript.

Kind regards, Eduardo José Lopes-Torres, Ph.D.Academic EditorPLOS Neglected Tropical Diseases  Jong-Yil ChaiSection EditorPLOS Neglected Tropical Diseases 

Shaden Kamhawi

co-Editor-in-Chief

Paul Brindley

co-Editor-in-Chief

**Journal Requirements:**

1) Please upload all main figures as separate Figure files in .tif or .eps format. For more information about how to convert and format your figure files please see our guidelines:

2) Some material included in your submission may be copyrighted. According to PLOSu2019s copyright policy, authors who use figures or other material (e.g., graphics, clipart, maps) from another author or copyright holder must demonstrate or obtain permission to publish this material under the Creative Commons Attribution 4.0 International (CC BY 4.0) License used by PLOS journals. Please closely review the details of PLOSu2019s copyright requirements here: PLOS Licenses and Copyright. If you need to request permissions from a copyright holder, you may use PLOS's Copyright Content Permission form.

Potential Copyright Issues:

- Figures 4, 7, and S1. Please confirm whether you drew the images / clip-art within the figure panels by hand. If you did not draw the images, please provide (a) a link to the source of the images or icons and their license / terms of use; or (b) written permission from the copyright holder to publish the images or icons under our CC BY 4.0 license. Alternatively, you may replace the images with open source alternatives. See these open source resources you may use to replace images / clip-art:

**Reviewers' comments:**

**Figure resubmission:** While revising your submission, please upload your figure files to the Preflight Analysis and Conversion Engine (PACE) digital diagnostic tool, https://pacev2.apexcovantage.com/. PACE helps ensure that figures meet PLOS requirements. To use PACE, you must first register as a user. Registration is free. Then, login and navigate to the UPLOAD tab, where you will find detailed instructions on how to use the tool. If you encounter any issues or have any questions when using PACE, please email PLOS at figures@plos.org. Please note that Supporting Information files do not need this step. If there are other versions of figure files still present in your submission file inventory at resubmission, please replace them with the PACE-processed versions.

**Reproducibility:** To enhance the reproducibility of your results, we recommend that authors of applicable studies deposit laboratory protocols in protocols.io, where a protocol can be assigned its own identifier (DOI) such that it can be cited independently in the future. Additionally, PLOS ONE offers an option to publish peer-reviewed clinical study protocols. Read more information on sharing protocols at https://plos.org/protocols?utm_medium=editorial-email&utm_source=authorletters&utm_campaign=protocols

---

## [Editor Report · Decision Letter 2]

2 Dec 2024

Dear Dr Bryant,

We are pleased to inform you that your manuscript 'Thermosensory behaviors of the free-living life stages of *Strongyloides* species support parasitism in tropical environments' has been provisionally accepted for publication in PLOS Neglected Tropical Diseases.

Best regards,

Eduardo José Lopes-Torres, Ph.D.

Academic Editor

Jong-Yil Chai

Section Editor

Shaden Kamhawi

co-Editor-in-Chief

Paul Brindley

co-Editor-in-Chief

---

## [Editor Report · Acceptance letter]

9 Dec 2024

Dear Dr Bryant,

We are delighted to inform you that your manuscript, "Thermosensory behaviors of the free-living life stages of *Strongyloides* species support parasitism in tropical environments," has been formally accepted for publication in PLOS Neglected Tropical Diseases.

Best regards,

Shaden Kamhawi

co-Editor-in-Chief

Paul Brindley

co-Editor-in-Chief
